# Clustering of Lifestyle Risk Factors among Algerian Adolescents: Comparison between Urban and Rural Areas: GSHS Data

**DOI:** 10.3390/ijerph18137072

**Published:** 2021-07-02

**Authors:** Abdelhamid Kerkadi, Hissa Al Mannai, Dana Saad, Fatima al Zahra Yakti, Grace Attieh, Hiba Bawadi

**Affiliations:** Human Nutrition Department, College of Health Sciences, QU-Health, Qatar University, Doha 2713, Qatar; ha1602298@student.qu.edu.qa (H.A.M.); ds1604970@student.qu.edu.qa (D.S.); fy1702033@student.qu.edu.qa (F.a.Z.Y.); gattieh@qu.edu.qa (G.A.); hbawadi@qu.edu.qa (H.B.)

**Keywords:** Algerian lifestyle, behavior risk factors, adolescents, GSH survey, urban, rural, clustering of LBRs

## Abstract

Objective: Compare the clustering of LBRs between urban and rural Algerian adolescents. Design: Data of this cross-sectional study was derived from the Global School-based Health Survey (GSHS). A self-administered, anonymous questionnaire was filled out by 4532 adolescents (11–16 years), which addressed LBRs of NCDs. Life style behavioral risk factors (LBRs) clustering was measured by the ratios of observed (O) and expected (E) prevalence of one or more simultaneously occurring LBRs for urban and rural areas separately. Multivariate logistic regression was performed to examine the association of LBRs as dependent variable with demographic variables (location, age, gender). Results: The most common LBR was physical inactivity (84.6%: 50.9% for urban and 49.1% for rural). Adolescents in urban areas had a higher prevalence of two (56.8% vs. 43.2%) and three and more (61.3% vs. 38.7%) LBRs than in rural areas. In urban areas, a significant positive association was found between (low fruit and vegetable consumption + physical inactivity) [2.06 (1.61–2.64)] and (high SB + smoking) [2.10 (1.54–2.76)], while (physical inactivity + high SB) [0.70 (0.54–0.91)] showed a significant negative association. In rural areas, (high SB + overweight/obesity) [1.49 (1.09–2.04)] had a significant positive association. While, (low fruit and vegetable consumption + high SB) [0.75 (0.60–0.94)], (physical inactivity + high SB) [0.65 (0.49–0.86)], and (physical inactivity + smoking) [0.70 (0.49–0.99)] had a negative association. Conclusions: Several socio-demographic factors have been identified to play a role in LBRs clustering among Algerian adolescents. Results of the study suggest the development of intervention aiming to tackle different LBRs rather than focusing on a single LBR.

## 1. Introduction

Non-communicable diseases (NCDs), such as cardiovascular diseases, diabetes, cancers, and chronic respiratory diseases, are considered the leading cause of death globally, contributing to 71% of the total deaths worldwide in 2016 [1]. Moreover, the prevalence of NCDs is increasing steadily in most of the Arab countries, and deaths related to NCDs and constitute 20% in third of the Arab countries, and 76% in Algeria [1,2].

Most of the NCDs patients develop the disease at adulthood, which may lead to early death in some cases [3]. Notably, many lifestyle patterns such as high screening time, physical inactivity, smoking, unhealthy eating patterns, and being overweight or obese play a major role in the development or prevention of these diseases [3].

A study reported that the prevalence of smoking among adolescents is higher in rural than urban areas, while sedentary behavior was more prevalent in urban than rural areas and the prevalence of obesity was 1.2% in urban and 1.4% in rural areas [4].

Clustering of NCDs behavioral risk factors occurs when the observed prevalence of co-occurrence of one or more risk factors exceeds their expected prevalence (O/E ratio >1) in individuals and populations [5]. Several studies assessed the prevalence of NCDs risk factors and their clustering among adolescents [5,6,7,8]. One study reported that low fruit and vegetable intake (86.9%) along with physical inactivity (58.6%) were the most prevalent risk factors for cardiovascular diseases (CVDs) among Bangladeshi adolescents [6]. Moreover, 17.6% of the adolescents had three or more risk factors [6]. Another study showed that low fruit and vegetable consumption and physical inactivity among adolescents were the most common risk factors for NCDs globally in both genders [5], while sedentary behavior was more likely to be found in adolescents who were smokers than non-smokers regardless of gender [5]. Additionally, a study showed that 80.6% of Brazilian adolescents had a low intake of fruits and vegetables which was the most common isolated risk factor, followed by physical inactivity (79.5%), alcohol consumption (23.8%), and tobacco use (5.6%) [8]. Highest proportion was for adolescents who had a cluster of two risk factors (56.1%). About 66% of Brazilian adolescents had a low consumption of fruits and vegetables and physical inactivity, which was the most prevalent risk factors cluster, and the aggregation of smoking and alcohol consumption (79.0%) had the highest prevalence of observed/expected ratio [8]. A study in Nepal showed that 82% of the adolescents had at least two risk factors and 11.2% had at least three [7]. The prevalence of risk factors clusters was higher among males than females in which 5.9% had inadequate fruit and vegetable intake and overweight/obesity, 4.8% had physical inactivity and overweight/obesity and 2.6% had smoking and alcohol [7]. We expect differences in lifestyle behavior risk factors between urban and rural areas in Algeria. To the best of our knowledge, no data exists to document the clustering lifestyle risk behaviors among Algerian adolescents. Therefore, the objectives of the present study were to assess the clustering of different lifestyle risk factors among Algerian adolescents and to compare it between urban and rural areas.

## 2. Materials and Methods

### 2.1. Data Source

Secondary data was used in this cross-sectional study, which was derived from the Global School-based Health Survey (GSHS) https://extranet.who.int/ncdsmicrodata/index.php/catalog/668 (accessed on 7 September 2020). In 2011, the survey was conducted by the World Health Organization (WHO) and United States Centers for Disease Control and Prevention [9]. The study was developed by using a self-administrated survey to estimate health behaviors for school-aged children aged 11–16 years [9]. The questionnaire addressed various topics such as demographics, dietary behaviors, physical activity, and sedentary behaviors, as well as tobacco and alcohol use that are identified as behavioral risk factors of NCDs. General methods and overall results of this survey have been previously published [9].

### 2.2. Study Participants

A self-administered and anonymous questionnaire was conducted among 4532 students consisting of 2150 males and 2326 females aged 11–16 years. Only 4189 participants were used in the data analysis out of 4532, 343 subjects with missing data on lifestyle were not included in the data analysis. Sample characteristics which include age and gender were assessed by asking the following questions: “How old are you?: 11 years old or younger, 12 years old, 13 years old, 14 years old, 15 years old, and 16 years old or older”. The other question was: “What is your sex?: Male and Female”.

### 2.3. Measurements

Different questions were asked to the survey participants to assess several lifestyle behaviors. To assess sedentary behavior among participants, the following question was asked: “How much time do you spend during a typical or usual day sitting and watching television, playing computer games, talking with friends, or doing other sitting activities such as surfing the internet?” with answer choices being <1, 1 to 2, 3 to 4, 5 to 6, 7 to 8, or >8 h/day. To assess fruits and vegetables consumption, “During the past 30 days, how many times per day did you usually eat fruit, such as bananas, apples, oranges, dates, or any other fruits?” and “During the past 30 days, how many times per day did you usually eat vegetables, such as potatoes, carrots, tomatoes, lettuce, or other vegetables?” questions were used with response choices being “I did not eat fruit during the past 30 days”; <1, 1, 2, 3, 4, or ≥5 times/day, and “I did not eat vegetables during the past 30 days”; < 1, 1, 2, 3, 4, or ≥5 times/day. To estimate the cigarette smoking rate among participants, the following question was asked “During the past 30 days, on how many days did you smoke cigarettes?” with the following response choices: 0, 1 or 2, 3 to 5, 6 to 9, 10 to 19, 20 to 29, or all 30 days. As the last lifestyle behavior, physical activity was estimated by asking: “During the past 7 days, on how many days were you physically active for a total of at least 60 min per day?” to participants with the following response choices: 0, 1, 2, 3, 4, 5, 6, 7 days.

### 2.4. Definition of Behavioral Risk Factors of NCDs

High sedentary behavior was determined as 3 h or more per day of sedentary time outside the school [10]. Physical inactivity was determined as adolescents not engaged in physical activity for at least 60 min per day of the week [10]. Adolescents were considered smokers if they smoked at least one day in the past 30 days [11]. Fruit and vegetable consumption was considered low if the intake of fruits and vegetables was less than two and three times per day in the past 30 days, respectively [12]. A dichotomous variable code (Yes = 1; No = 0) was given to each risk behavior to be added to generate a risk factor index that ranges from 0 (no risk factor) to 5 (all risk factors).

### 2.5. Anthropometric Data

Body mass index was calculated as body weight in kilograms divided by height in meters squared, and was classified into three categories from the median BMI for age and sex: underweight if BMI was <−2 standard deviation (SD), overweight if BMI was >+1SD, and obese if BMI was >+2SD, according to the WHO Child Growth Standards [13].

### 2.6. Statistical Analysis

Data were analyzed using SPSS statistical package version 23. Descriptive statistics are presented as mean, standard deviation, and proportion. Differences between sex and location (rural and urban) were determined by the Student *t* test for continuous variables, and ƛ2 test for categorical data.

The clustering of lifestyle factors among adolescents was measured by the ratios of observed (O) and expected (E) prevalence of one or more simultaneously occurring risk behaviors for urban and rural areas separately. The presence of clustering was identified if the observed/expected ratio (O/E) was higher than one [5].

Multivariate logistic regression was performed to examine the association of lifestyle risk factors as dependent variable with demographic variables such as location, sex, and age. The significance of the differences was set at the level of *p* < 0.05.

## 3. Results

The study included 4189 Algerian adolescents living in urban (*n* = 2131) and rural areas (*n =* 2058). A total of 47.7% were male with no significant difference in sex distribution between rural and urban areas. As shown in Table 1, almost 85% of adolescents were aged 13 and above with a significant difference in age group distribution between locations (*p* < 0.001). Adolescents living in the urban area were significantly heavier than their peers in the rural area (*p* = 0.001) presenting a significantly higher BMI (*p* < 0.001). The prevalence of underweight, overweight, and obesity was 6.8%, 13.7%, and 3.2%, respectively. We noted a significant difference in the prevalence of underweight and overweight between locations. Both overweight and obesity were more common in urban area (58.1% vs. 41.9% for overweight and 59.3% vs. 40.7% for obesity). In contrast to overweight and obesity, underweight was significantly higher (*p* = 0.004) in the rural area (57.4% vs. 42.6%). There was no significant difference in height according to location.

Table 2 shows that the most common single behavioral risk factor was physical inactivity, with a prevalence of 84.6% (50.9% for urban and 49.1% for rural). Low fruit and vegetable intake was the second most common risk behavior with a prevalence of 66.8% (49.9% for urban and 50.1% for rural). A total of 13.2% of adolescents were either overweight or obese with a significantly higher rate in the urban area (57.3% vs. 42.7%, *p* = 0.001). We noted that 26.5% of adolescents were adopting a sedentary behavior and urban adolescents were more likely to report high sedentary behavior (59% vs. 41%, *p* < 0.0001). Only 10.7% of adolescents reported using tobacco and there were no significant differences between rural and urban area. For the total number of LBR, 34.9% of adolescents did not have any LBR. While 38.8% of adolescents have one LBR, 19.8% of adolescents had two LBRs and 6.5% had three or more LBRs. Compared to the rural area, the urban area had a higher prevalence of two and three and more LBRs, respectively, while the rural area had a higher rate of zero and one LBR, respectively.

Table 3 shows the clustering effects of the five health risk factors, categorized by location. In general, the O/E ratios among different combinations of the five health risk factors were higher in urban population. However, the O/E ratio of having no health risk behaviors was higher among rural population. Among the 32 combinations of the five health risk factors, the clustering effect was shown in 14 combinations in the urban population, while only six combinations revealed a clustering effect in rural population. The strongest across the 32 combinations was shown in urban population for the clustering of (high ST + OW/OB + smoking) and the clustering of (high ST + low PA + OW/OB + smoking) with an O/E ratio of 1.50 for both. In rural population, the highest O/E ratio of 1.27 was for the clustering of low PA and smoking. The clustering of any of the risk behaviors with both low PA and smoking together was highly observed in urban and rural populations. This was evident in the clustering of high ST, low PA, OB/OW, and smoking that had the strongest O/E ratio (1.50) among urban population, and clustering of low PA and smoking that had the strongest O/E ratio (1.27) among rural population.

Table 4 shows the association between number of LBRs and demographic indicators. Results indicated that adolescents living in the urban area were less likely to have one and two LBRs (OR 0.605 [95% CI 0.413–0.884]); (0.639 [0.442–0.924]) for one and two LBRs, respectively. In contrast, no association was found between geographical location and having three or more lifestyle behavior risks. There was no association between having one or two lifestyle behavior risks and age group, whereas the youngest (OR 0.561 [95% CI 0.322–0.977]) and middle age groups (0.586 [0.390–0.879]) were significantly less likely to have three or more lifestyle behavior risks, compared to the old group (*p* < 0.05). According to gender, males were less likely to have one (OR 0.424 [95% CI 0.286–0.630]), two (0.411 [0.280–0.604]), and three or more lifestyle behavior risks (0.517 [0.349–0.766]) compared to females.

Table 5 shows the possible association between the five health risk factors, categorized by location, using logistic regression, adjusted for age and gender. Among the 10 pairs of clustering in the urban population, a significant association was shown between three pairs where two pairs (low fruit and vegetable consumption + physical inactivity) and (high SB + smoking) had a positive association and one pair (physical inactivity + high SB) had a negative association. On the other hand, among the 10 pairs in the rural population, a significant association was indicated between four pairs, with one pair being positively associated (high SB + overweight/obesity) while three pairs being negatively associated (low fruit and vegetable consumption + high SB), (physical inactivity + high SB), and (physical inactivity + smoking). Among both urban and rural populations, physical inactivity was negatively associated with high SB (OR 0.70 [95% CI 0.54–0.91]) and (0.65 [0.49–0.86]), respectively. In the urban population, low intake of fruits and vegetables was positively associated with physical inactivity (OR 2.06 [95% CI 1.61–2.64]), and a high SB was positively associated with smoking (2.10 [1.54–2.76]). In the rural population, low intake of fruits and vegetables was negatively associated with a high SB (OR 0.75 [95% CI 0.60–0.94]), and physical inactivity was negatively associated with smoking (0.70 [0.49–0.99]). Moreover, a high SB was positively associated with overweight/obesity (OR 1.49 [95% CI 1.09–2.04]) in rural population. Despite of having a positive association between smoking and overweight/obesity in the general population, there was no significant association between them in the categorized populations.

## 4. Discussion

To the best of our knowledge, this study is the first to prove the clustering pattern of NCDs risk factors among Algerian adolescents aged 11–16 years, and it is based on data from a nationally representative survey. Five lifestyle behavioral risk factors including high sedentary behavior, low physical activity, low fruit and vegetable consumption, smoking, and overweight/obese were studied. Our results investigate the burden of NCDs risk factors among urban and rural Algerian adolescents.

The most common single behavioral risk factors among Algerians were physical inactivity and low fruit and vegetable intake, with a prevalence of 84.6% (50.9% for urban and 49.1% for rural), and 66.8% (49.9% for urban and 50.1% for rural), respectively. Compared to other studies, the results were similar, such as in Malaysia, where 85.5% and 70.4% of adolescents were physically inactive and had a low consumption of fruits and vegetables, respectively [14]. Most of Chinese adolescents (92.8%) did not meet physical activity recommendations also, especially in peri-urban areas [15]. In the UK, 73.6% of the adolescents had insufficient fruits and vegetables consumption [16]. Furthermore, 95.6% presented low fruit consumption and 79.9% presented low consumption of vegetables in Brazil [17]. The prevalence of physical inactivity in Nepal (84.77%) was similar to the one in Algeria, while insufficient consumption of fruits and vegetables was the most prevalent risk factor (95.33%) [7].

Other factors which were most prevalent in other studies were sedentary behavior and unhealthy food consumption (Nunes et al., 2016; Shayo, 2019). An unhealthy diet was consumed by 92.1% and 80.5% of Brazilian [18] and Tanzanian [19] adolescents, respectively. Sedentary behavior was the second most prevalent risk adopted by Brazilian adolescents (87.4%), which was followed by physical inactivity (77.3%) [18].

In contrast to Algerians, rural Swedish adolescents were more physically inactive compared to urban (42% vs. 35.1%) [20], while in Brazil, results indicated that adolescents living in rural areas were more physically inactive compared to adolescents from urban areas (69.2% vs. 63.4%) [21]. Similar findings to our study, a low fruit and vegetable consumption rate among rural adolescents was reported by Ricardo et al. in Brazil, while in Sweden, the rate of low consumption of fruits and vegetables did not significantly differ between rural and urban area, respectively (41.8% vs. 40.3%) [20]. A total of 26.5% of Algerian adolescents were adopting a sedentary behavior, especially in urban areas. Similarly, an Indonesian study reported that sedentary behavior was more prevalent in urban areas [22].

In Algeria, 34.9% of adolescents did not have any LBR, 38.8% had one LBR, 19.8% had two LBRs, and 6.5% had three or more LBRs. Having none or one risk factor was also mostly prevalent in the UK (none or one; 30%) [16]. In contrast to our findings, having two LBRs was mostly prevalent among Brazilian (38.0%) [21] and Malaysian (39.7%) [14] adolescents. In another Brazilian study, almost half (48.5%) of adolescents had three LBRs [18].

Urban adolescents had a higher prevalence of two and three and more LBRs, while adolescents in rural areas had a higher rate of zero and one LBR in Algeria. In Brazil, adolescents living in urban areas had a higher prevalence of four or more LBRs, which shows an increase in LBR number in urban areas [21].

In Algeria, younger adolescents (11–14 years old) were significantly less likely to have three or more LBRs, compared to the eldest group (15+ years old). This could be explained by an increase in number of LBR with age. Similar findings were shown in a global [5] and Brazilian [21] study, where the number of LBR increased with age. Moreover, Nepalese adolescents aged 17 years old were among the highest age groups to have the highest number of clustered risk factors [7]. However, older Tanzanian adolescents had a lower number of LBR than the younger group [19]. Another Brazilian study showed no significant difference between the number of LBR and age [18].

In Algeria, males were significantly less to have one, two, and three or more risk behaviors. Our results are in line with other studies. In Tanzania, male adolescents were significant less likely to have one, two, four, and five LBRs than females [19]. Furthermore, 4 > LBRs were more prevalent among Brazilian females also [21]. Another Brazilian study did not find a significant association between the number of LBR and gender [18]. In Bangladesh, males were more frequent to have three or more (21.3% vs. 11.2%) risk behaviors [6]. Moreover, risk factors were more likely to cluster in Nepalese males [7].

The highest clustering among Algerian adolescents was low fruit and vegetable consumption + high sedentary behavior. In Tanzania, the most prevalent cluster was for unhealthy diet + physical inactivity [19]. The second most prevalent cluster in Algeria was low fruit and vegetable consumption + low physical activity. Similarly, in Brazil [8], in the global study [5], and in Nepal [7], insufficient physical activity + low consumption of fruits and vegetables was the most prevalent cluster among adolescents. In Brazil, physical inactivity, unhealthy diet, and sedentary behavior was found to be the most prevalent cluster adopted by adolescents (40.1%) [18].

In the studied urban population, low intake of fruits and vegetables was positively associated with physical inactivity. The same findings were noticed in Brazil [17], Bangladesh [6], and among males in the global study [5], were those who were physically active had more consumption of fruits and vegetables. Moreover, in Algerian urban areas, there was a significant positive association between high SB and smoking. The same findings were reported in the global study [5]. In the rural Algerian population, low intake of fruits and vegetables was negatively associated with high SB contradicting Bangladeshi females, who had a positive association [6].

Other studies have found positive associations between different LBRs such as smoking and alcohol [5,6]. In a global study, smoking was positively associated with alcohol consumption [5]. Furthermore, alcohol consumption was positively associated with physical inactivity, high sedentary behavior, and smoking among Bangladeshi males [6].

Findings from the present study shows that behavioral risk factors do not occur in isolation and differ according to the living areas. This indicates a need for a comprehensive approach that takes into consideration the clustering of these behaviors and the difference in their prevalence in urban and rural areas, to promote health behaviors and reduce NCDs in adulthood. This study highlights the importance of identifying LBRs associated with NCDs among adolescents to develop prevention programs that can help in decreasing the rate of NCDs in adulthood [3]. By identifying the most common LBRs and their clustering among adolescents, as they are in an early stage of life, preventive interventions can be applied in schools and society targeting adolescents to minimize the prevalence of NCDs among adults. Implications from this study for public health include implementing activities for disease prevention and life-long health offers which can help prevent or at least delay the onset of these NCDs.

The current study is the first among Algerian adolescents to document the clustering of LBRs. In addition, the use of a large nationally representative number of adolescents enrolled in school in rural and urban area. However, it has some limitations. The GSHS includes only adolescents attending schools, which may not be representative of the total adolescent population. All LBRs have been assessed using a self-administered questionnaire, which may affect the accuracy of our results. Other important factors were not collected such as socio-economic data and anthropometric measurements were self-reported. The cross-sectional study design did not allow to establish a causal relationship between clustering and location.

## 5. Conclusions

Findings from the present study highlighted the high prevalence of multiple lifestyle behavioral risk factors among Algerian adolescents. The co-existence of LBRs was present more often than expected. There was a significant difference in the prevalence of LBRs between rural and urban adolescents. Several socio-demographic factors (location, age, gender) have been identified to play a role in the LBRs clustering among adolescents in Algeria. Further researches including objective measures of LBRs, socio-economic status indicators, and blood biomarkers are required to understand the importance of LBRs clustering in the development of NCDs among adolescents. Results of the present study suggest the development of intervention aiming to tackle different LBRs rather than focusing on strategies that address a single LBR. By identifying the most common LBRs and their clustering among adolescents, as they are in an early stage of life, preventive interventions can be applied in schools and society targeting adolescents to minimize the prevalence of NCDs among adults. Implications from this study for public health include implementing activities for disease prevention such as an increase awareness of the importance of a healthy life style (high physical activity, healthy dietary pattern, and a low sedentary behaviors) that can help preventing or at least delaying the onset of these NCDs.

## Figures and Tables

**Table 1 ijerph-18-07072-t001:** Descriptive statistics of the study population according to location (urban/rural).

	Total (*n* = 4189)	Urban(*n* = 2131)	Rural (*n* = 2058)	*p*-Value
Demographic data Gender Male N (%) Female N(%) Age (years) G1 (11–12 y/o) N(%) G2 (13–14 y/o) N (%) G3 (15+ y/o) N (%)	1999 (47.7%) 2190 (52.3%) 602 (14.4%) 1709 (40.8%) 1878 (44.8%)	999 (50%) 1132 (51.7%) 338 (56.1%) 808 (47.3%) 985 (52.4%)	1000 (50%) 1058 (48.3%) 264 (43.9%) 901 (52.7%) 893 (47.6%)	0.268 <0.001
Anthropometric data Weight (kg) Height (m) BMI (kg/m^2^) Underweight N (%) Normal weight N(%) Overweight N (%) Obese N (%)	Mean (SD) 48.8 (10.6) 1.57 (0.10) 19.4 (3.2) 284 (6.8%) 3197 (76.3%) 573 (13.7%) 135 (3.2%)	Mean (SD) 49.3 (10.7) 1.57 (0.10) 19.7 (3.2) 21 (42.6%) 1597 (49.9%) 333 (58.1%) 80 (59.3%)	Mean (SD) 48.1 (10.5) 1.58 (0.09) 19.1 (3.2) 163 (57.4%) 1600 (50.1%) 240 (41.9%) 55 (40.7%)	0.001 0.186 <0.001 0.004 0.878 <0.001 0.054

BMI: body mass index, G: age group, underweight = BMI < −2SD, overweight = BMI > +1SD, obese = BMI > +2SD. Values for weight, height, and BMI are expressed as mean (SD).

**Table 2 ijerph-18-07072-t002:** Prevalence of lifestyle risk factors of Algerian adolescents aged 11–16 years.

Risk Factors	Urban *n* = 2130	Rural *n* = 2058	Total *n* = 4188	*p*-Value
Overweight/Obesity N (%)	413 (58.3%)	295 (41.7%)	708 (16.9%)	0.001
High sedentary behavior N (%)	654 (59.0%)	455 (41.0%)	1109 (26.5%)	<0.001
Low fruit & vegetable intake N (%)	1397 (49.9%)	1403 (50.1%)	2800 (66.8%)	0.072
Smoking N (%)	246 (54.7%)	204 (45.3%)	450 (10.7%)	0.089
Physical inactivity N (%)	1803 (50.9%)	1742 (49.1%)	3545 (84.6%)	0.973
Number of LBR N (%)				<0.001
0	683 (46.7%)	778 (53.3%)	1461 (34.9%)	
1	811 (49.8%)	816 (50.2%)	1627 (38.8%)	
2	471 (56.8%)	358 (43.2%)	829 (19.8%)	
3+	166 (61.3%)	105 (38.7%)	271 (6.5%)	

Low F/V = ate less than 5 F/V/day, physical inactivity = less than 60 min PA/day, high SB = sitting >3 hr/day, overweight = BMI > +1SD, obese = BMI > +2SD, smoking = smoked at least one day last month, LBR: lifestyle behavior risk factor.

**Table 3 ijerph-18-07072-t003:** The clustering effect of the five lifestyle risk behaviors among Algerian adolescents aged 11–16 years, Global School-based Student Health Survey, Algeria, 2011.

Number of LBR	High ST	Low PA	Low F/V	OW/OB	Smoking	Urban			Rural			
						n	O	E	O/E	O	E	O/E
0	-	-	-	-	-	1495	33.0	35.7	0.92	38.5	35.7	1.08
1	-	-	-	-	+	143	3.3	3.5	0.94	3.5	3.4	1.03
1	-	-	-	+	-	214	5.2	5.1	1.02	5.1	5.1	1.00
1	-	-	+	-	-	690	15.3	16.5	0.93	17.6	16.5	1.07
1	-	+	-	-	-	201	3.8	4.8	0.79	5.8	4.8	1.20
1	+	-	-	-	-	389	10.6	9.3	1.14	8.0	9.3	0.86
2	+	+	-	-	-	83	2.0	2.0	1.00	2.0	2.0	1.00
2	+	-	+	-	-	218	5.7	5.2	1.01	4.7	5.2	0.90
2	+	-	-	+	-	91	2.8	2.2	1.27	1.5	2.2	0.68
2	+	-	-	-	+	76	2.4	1.8	1.33	1.2	1.8	0.67
2	-	+	+	-	-	118	3.6	2.8	1.29	2.0	2.8	0.71
2	-	+	-	+	-	14	0.3	0.3	1.00	0.3	0.3	1.00
2	-	+	-	-	+	48	0.9	1.1	0.82	1.4	1.1	1.27
2	-	-	+	+	-	103	2.7	2.5	1.08	2.2	2.5	0.88
2	-	-	+	-	+	41	0.8	1.0	0.80	1.1	1.0	1.10
2	-	-	-	+	+	20	0.5	0.5	1.00	0.5	0.5	1.00
3	+	+	+	-	-	66	1.8	1.6	1.13	1.3	1.6	0.81
3	+	+	-	+	-	7	0.2	0.2	1.00	0.1	0.2	0.50
3	+	+	-	-	+	17	0.4	0.4	1.00	0.4	0.4	1.00
3	+	-	+	+	-	38	0.9	0.9	1.00	0.9	0.9	1.00
3	+	-	+	-	+	30	1.0	0.7	1.43	0.4	0.7	0.60
3	+	-	-	+	+	8	0.3	0.2	1.50	0.0	0.2	0.00
3	-	+	+	+	-	11	0.4	0.3	1.33	0.1	0.3	0.33
3	-	+	+	-	+	20	0.5	0.5	1.00	0.5	0.5	1.00
3	-	-	+	+	+	11	0.4	0.3	1.30	0.1	0.3	0.33
3	-	+	-	+	+	2	0.0	0.0	0.00	0.0	0.0	0.00
4	+	+	+	+	-	2	0.0	0.0	0.00	0.0	0.0	0.00
4	-	+	+	+	+	2	0.0	0.0	0.00	0.0	0.0	0.00
4	+	-	+	+	+	6	0.1	0.1	1.00	0.1	0.1	1.00
4	+	+	-	+	+	7	0.3	0.2	1.50	0.0	0.2	0.00
4	+	+	+	-	+	14	0.4	0.3	1.33	0.3	0.3	1.00
5	+	+	+	+	+	3	0.1	0.1	1.00	0.0	0.1	0.00

PA: physical activity, ST: screen time, F/V: fruits and vegetables, OW/OB: overweight and obesity, O: Observed prevalence, E: expected prevalence, O/E; ratio between observed and expected prevalence, RBH: risk behaviors, (+) sign indicates presence of a risk factor and (-) sign indicates absence of a risk factor, low F/V = ate less than 5 F/V/day, low PA = less than 60 min PA/day, high ST = sitting >3 hr/day, overweight = BMI > +1SD, obese = BMI > +2SD, smoking = smoked at least one day last month.

**Table 4 ijerph-18-07072-t004:** Multinomial logistic regression analysis of factors associated with combination of lifestyle behavior risk in Algerian adolescents.

Variables	One LBR OR (95% CI)	Two LBR OR (95% CI)	3 and + LBR OR (95% CI)
Geographical location			
Urban	0.605 (0.413–0.884) *	0.639 (0.442–0.924) *	0.908 (0.622–1.326)
Rural	1	1	1
Age groups (years)			
11–12	0.990 (0.570–1.721)	0.789 (0.460–1.352)	0.561 (0.322–0.977) *
13–14	0.903 (0.599–1.359)	0.837 (0.563–1.245) *	0.586 (0.390–0.879) *
15+	1	1	1
Gender			
Male	0.424 (0.286–0.630) ***	0.411 (0.280–0.604) ***	0.517 (0.349–0.766) **
Female	1	1	1

LBR: lifestyle behavior risk factor, OR: odds ratio, 95% CI: 95% confidence interval, * *p* < 0.05; ** *p* < 0.01; *** *p* < 0.001.

**Table 5 ijerph-18-07072-t005:** Odds ratio (95% CI) for the co-occurrence of two lifestyle behavioral risk factors in Algerian adolescents aged 11–16 years, stratified by location, Global School-based Student Health Survey, Algeria, 2011.

	Low F/V	Low PA	High SB	Overweight /Obesity	Smoking
General population: Low fruit and vegetable Physical inactivity High SB Overweight/obesity Smoking	1 **0.63 (0.53–0.75)** **0.86 (0.74–0.99)** 0.99 (0.81–1.19) 1.14 (0.91–1.43)	1 **0.69 (0.57–0.84)** 1.06 (0.81–1.40) 0.80 (0.62–1.02)	1 **1.30 (1.07–1.59)** **1.75 (1.41–2.18)**	1 **1.37 (1.00–1.87)**	1
Urban Low fruit and vegetable Physical inactivity High SB Overweight/obesity Smoking	1 **2.06 (1.61–2.64)** 0.96 (0.79–1.17) 1.01 (0.78–1.30) 1.14 (0.83–1.55)	1 **0.70 (0.54–0.91)** 0.94 (0.66–1.34) 0.88 (0.62–1.26)	1 1.14 (0.88–1.45) **2.10 (1.54–2.76)**	1 1.44 (0.96–2.16)	1
Rural Low fruit and vegetable Physical inactivity High SB Overweight/obesity Smoking	1 1.20 (0.91–1.56) **0.75 (0.60–0.94)** 0.99 (0.74–1.32) 1.13 (0.81–1.58)	1 **0.65 (0.49–0.86)** 1.27 (0.81–1.98) **0.70 (0.49–0.99)**	1 **1.49 (1.09–2.04)** 1.38 (0.99–1.93)	1 1.24 (0.75–2.05)	1

PA: physical activity, SB: sedentary behavior, F/V: fruits and vegetables, OW/OB: overweight and obesity, CI: confidence intervals, low F/V = ate less than 5 F/V/day, low PA = less than 60 min PA/day, high SB = sitting >3 hr/day, overweight = BMI > +1SD, obese = BMI > +2SD, smoking = smoked at least one day last month. Adjusted for age and gender. Significant associations are highlighted in bold.

## Data Availability

Datasets generated during the study can be obtained directly from the corresponding author at abdel.hamid@qu.edu.qa.

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
