# Peer review of "Clustering of Lifestyle Risk Factors among Algerian Adolescents: Comparison between Urban and Rural Areas: GSHS Data"

_ijerph, 2021, doi:10.3390/ijerph18137072_

Round 1

Reviewer 1 Report

Thank you for inviting me to review the paper: Clustering of lifestyle risk factors among Algerian adolescents : 2 comparison between urban and rural area: GSHS data.

The paper is well written, easily accessible and enjoyable to read. 

The introduction gives the reader the right context to understand the research questions. 

Methods are sound and clear. Tables are clearly presented.

Conclusion are derived from data and refer to other international findings.

A minor improvement from a reader perspective: I would like to understand more about the ethical standard used in the study to collect the information and in which way the results are going to be used for the target population.

Author Response

June 21st, 2021

Dear Editor-in-Chief

Many thanks for your recent message concerning our revised original manuscript. Encouraged by the positive opinion on the significance of our work by the Reviewers, we are now pleased to submit a revised version of this review manuscript. We have addressed (see below) the comments raised by the reviewers’ point-by-point. The changes are highlighted in the new version of the manuscript.

We hope you can now find the revised version of the manuscript suitable for publication in International Journal of Environmental Research and Public Health.

Yours truly,

Abdelhamid Kerkadi, PhD

Reviewer #1

Thank you for inviting me to review the paper: Clustering of lifestyle risk factors among Algerian adolescents : 2 comparison between urban and rural area: GSHS data.

The paper is well written, easily accessible and enjoyable to read. 

The introduction gives the reader the right context to understand the research questions. 

Methods are sound and clear. Tables are clearly presented.

Conclusion are derived from data and refer to other international findings.

A minor improvement from a reader perspective: I would like to understand more about the ethical standard used in the study to collect the information and in which way the results are going to be used for the target population.

 Response: I would like to thank the reviewer for the positive comments. The data was collected according to a protocol developed by the World Health Organization and all ethical approvals were obtained from each country where the survey was conducted. Results of the study can be used to develop awareness campaigns to reduce the risk of non communicable diseases among adolescents especially in the urban area.

Reviewer 2 Report

Dear authors
Congratulations on your work. I make some considerations that can improve the document.
1. In the Materials and Methods Data Sources section, the link to the World School Health Survey (line 72) does not work.
2. There is an error in tables 1 and 2. The percentages are incorrectly placed in the case of rural and urban, since they are placed by rows instead of by columns and the result is therefore not correct.
In total if it is correct. Percentages must be changed based on location.
3. For study participants, mean age can be indicated.
4.In Table 1, in the anthropometric data, the words Mean and Standard Deviation should be indicated in a row, since it is not specified what the data refer to, for example: 49.3 (10.7) and the others .
5. In table 1 it may be interesting to add the normal weight.
6. In table 2, the data does not come out considering table 1.
Overweight / Obesity N (%) 317 (57.3%) 236 (42.7%) 553 (13.2%) 0.001 The
P value is not correct
7. In Table 3, the label "urban" is misplaced.
8. Taking into account the suggestions in Tables 1 and 2, the other tables should be reviewed to verify that the data is correct.
9. It is recommended to include in the conclusions some suggestions to reduce risk factors in adolescence.
10. in reference 2, check the colon after springer
Sincerely

Author Response

June 21st, 2021

Dear Editor-in-Chief 
Many thanks for your recent message concerning our revised original manuscript. Encouraged by the positive opinion on the significance of our work by the Reviewers, we are now pleased to submit a revised version of this review manuscript. We have addressed (see below) the comments raised by the reviewers’ point-by-point. The changes are highlighted in the new version of the manuscript. 
We hope you can now find the revised version of the manuscript suitable for publication in International Journal of Environmental Research and Public Health.
Yours truly,
Abdelhamid Kerkadi, PhD

Reviewer #2

Dear authors

1. In the Materials and Methods Data Sources section, the link to the World School Health Survey (line 72) does not work.
Response: The link was added 
2. There is an error in tables 1 and 2. The percentages are incorrectly placed in the case of rural and urban, since they are placed by rows instead of by columns and the result is therefore not correct.
In total if it is correct. Percentages must be changed based on location.
Response : Data in table 1 and 2 were  revised and corrected.
3. For study participants, mean age can be indicated.
4.In Table 1, in the anthropometric data, the words Mean and Standard Deviation should be indicated in a row, since it is not specified what the data refer to, for example: 49.3 (10.7) and the others .
Response: data in table 1 was adjusted 
5. In table 1 it may be interesting to add the normal weight.
Response: The prevalence of normal weight was added in table 1
6. In table 2, the data does not come out considering table 1.
Overweight / Obesity N (%) 317 (57.3%) 236 (42.7%) 553 (13.2%) 0.001 The
P value is not correct
Response: table 2 was revisited and changed accordingly 
7. In Table 3, the label "urban" is misplaced.
Response: The label Urban was placed correctly  
8. Taking into account the suggestions in Tables 1 and 2, the other tables should be reviewed to verify that the data is correct.
Response : Data in table 3 and 4 were reviewed 
9. It is recommended to include in the conclusions some suggestions to reduce risk factors in adolescence.
Response: Suggestions to reduce the risk factors was added in the conclusion 
10. in reference 2, check the colon after springer
Response: Colon was removed in reference 2 

Reviewer 3 Report

Thank you for the opportunity to review the manuscript “Clustering of lifestyle risk factors among Algerian adolescents: comparison between urban and rural area: GSHS data” for International Journal of Environmental Research and Public Health. Thank you for conducting such an important study. Overall, it was an enjoyable read. Frankly, I’m impressed. The literature is well reviewed, the methods are clearly described, the findings and conclusions are well grounded, and limitations are acknowledged. Generally speaking, I have only favorable comments to offer. As such, I don’t want to slow down the review process asking for tangentially related revisions. So, instead, I will just say “well done.”

Author Response

June 21st, 2021

Dear Editor-in-Chief

Many thanks for your recent message concerning our revised original manuscript. Encouraged by the positive opinion on the significance of our work by the Reviewers, we are now pleased to submit a revised version of this review manuscript. We have addressed (see below) the comments raised by the reviewers’ point-by-point. The changes are highlighted in the new version of the manuscript.

We hope you can now find the revised version of the manuscript suitable for publication in International Journal of Environmental Research and Public Health.

Yours truly,

Abdelhamid Kerkadi, PhD

Reviewer #3

Thank you for the opportunity to review the manuscript “Clustering of lifestyle risk factors among Algerian adolescents: comparison between urban and rural area: GSHS data” for International Journal of Environmental Research and Public Health. Thank you for conducting such an important study. Overall, it was an enjoyable read. Frankly, I’m impressed. The literature is well reviewed, the methods are clearly described, the findings and conclusions are well grounded, and limitations are acknowledged. Generally speaking, I have only favorable comments to offer. As such, I don’t want to slow down the review process asking for tangentially related revisions. So, instead, I will just say “well done.”

Response: The authors appreciate the valuable comments of the reviewer

Reviewer 4 Report

The topic and the aim of the research is important for public health. It is an interesting and valuable study.

There are my comments that I believe can improve the quality of the manuscript.

Line 9, p. 1. Abbreviations in abstract should be clarified. What is LBRs? GSHS? Abbreviation LBRs is never presented in following text as well.

Introduction presents data of the health – related lifestyle in various countries. It would be interesting to have some data on health -related  lifestyle of Algerian adolescents in general to understand the tendencies of lifestyle changes.  

The aim of the study might be revised. I suggest – the aim of the study was to assess health – related lifestyle of Algerian adolescents and to compare it between urban and rural adolescents.  

Line 8, p. 2 – what question was for dividing adolescents to urban and rural?

Table 1. I don‘t see rationale to name underweight, overweight and obese groups as G1, G2 and G3. Please, name it as „underweight, overweight and obese“.

Discussion. It is important to understand if data from the present study differs from previous lifestyle data of Algerian adolescents? If possible, please include it.

Line 254, p. 8  - referencing (Nunes, et al. 2016; Shayo, 2019) is not correct.

Author Response

June 21st, 2021

Dear Editor-in-Chief

Many thanks for your recent message concerning our revised original manuscript. Encouraged by the positive opinion on the significance of our work by the Reviewers, we are now pleased to submit a revised version of this review manuscript. We have addressed (see below) the comments raised by the reviewers’ point-by-point. The changes are highlighted in the new version of the manuscript.

We hope you can now find the revised version of the manuscript suitable for publication in International Journal of Environmental Research and Public Health.

Yours truly,

Abdelhamid Kerkadi, PhD

Reviewer #4

The topic and the aim of the research is important for public health. It is an interesting and valuable study.

  1. There are my comments that I believe can improve the quality of the manuscript.
  2. Line 9, p. 1. Abbreviations in abstract should be clarified. What is LBRs? GSHS? Abbreviation LBRs is never presented in following text as well.

Response: The abbreviations were clarified in the revised manuscript

  1. Introduction presents data of the health – related lifestyle in various countries. It would be interesting to have some data on health -related  lifestyle of Algerian adolescents in general to understand the tendencies of lifestyle changes.  

Response : To my knowledge , there was no recent data published on health life style of Algerian adolescents.

  1. The aim of the study might be revised. I suggest – the aim of the study was to assess health – related lifestyle of Algerian adolescents and to compare it between urban and rural adolescents.

Response: The aim was revised and adjusted accordingly

  1. Line 8, p. 2 – what question was for dividing adolescents to urban and rural?

Response: we divided the study population into  urban and rural to elucidate the difference in LBRs between them . Knowing that life conditions are different between urban and rural areas in Algeria.

  1. Table 1. I don‘t see rationale to name underweight, overweight and obese groups as G1, G2 and G3. Please, name it as „underweight, overweight and obese“.

Response: The G1, G2 and G3 were for age groups and not BMI groups. Footnote was corrected

  1. It is important to understand if data from the present study differs from previous lifestyle data of Algerian adolescents? If possible, please include it.

Response: The authors agreed  with the reviewer comments. Unfortunately, most published data on Algerian adolescents were not representative of the total Algerian population. They were conducted among adolescents living in specific region (East or West)  and they did not include life style factors. Most of them covered the magnitude of obesity, underweight and overweight.

  1. Line 254, p. 8  - referencing (Nunes, et al. 2016; Shayo, 2019) is not correct.

Response: The references were revised and changed in the text ( highlighted in red)
